# POLYGLOT-R1: REINFORCEMENT LEARNING FOR MULTILINGUAL MULTI-PERSPECTIVE REASONING

## ABSTRACT

Multilingual reasoning has recently emerged as a powerful strategy for extending the reach and impact of large language models (LLMs). By enabling models to operate effectively across diverse languages and modalities, it broadens access to advanced reasoning capabilities for a wider range of users and linguistic communities. Yet reliably activating such behaviours through training remains difficult. Existing approaches rely heavily on supervised fine-tuning over synthetic data, which tends to encourage imitation of teacher signals rather than genuine exploration or robust generalisation. To address this gap, we introduce, we propose **Polyglot-R1**, the first reinforcement learning framework designed to cultivate multilingual, multi-perspective reasoning behaviours for complex, real-world tasks. Our framework introduces a progressive curriculum that directly tackles the cold-start problem in training with reinforcement learning. We begin with supervised fine-tuning on trajectories from more straightforward multilingual prompts to instil the foundations of this reasoning style. We then transition to reinforcement learning, enabling the model to actively explore and generalise this skill on more challenging multilingual and multimodal problems. Experiments demonstrate that Polyglot-R1 not only improves accuracy but also reshapes the way models reason. At earlier stages of training, multilingual reasoning functions as an exploration strategy, encouraging the model to test diverse lines of thought. At later stages, the same capacity is repurposed as a mechanism for multi-perspective verification, strengthening confidence in the final answer. Most importantly, we validate multilingual reasoning as an intermediate exploration scaffold: a temporary but crucial phase that unlocks more robust, transferable reasoning capabilities across languages.

## 1 INTRODUCTION

Large language models (LLMs) are increasingly deployed in multilingual contexts, where their reasoning capabilities influence how knowledge is accessed across cultural and linguistic boundaries. Yet their reasoning often remains brittle, overly sequential, and skewed towards high-resource languages. This reveals a gap between surface-level fluency and the deeper, transferable reasoning skills required for truthfulness, reliability, and inclusivity.

Recent research highlights the promise of multi-path reasoning strategies—*such as parallel thinking*—which enable models to explore alternative lines of thought before synthesising them into a coherent conclusion Luong & Lockhart (2025); Zheng et al. (2025). Cognitive science suggests that humans employ similar strategies to avoid premature commitment to a single, potentially flawed solution (Clark, 1989; Jackendoff, 2011). Yet most methods for instilling such capacities in LLMs are limited. Supervised fine-tuning on synthetic traces tends to encourage imitation rather than genuine exploration, while test-time prompting increases inference costs without delivering lasting improvements. Reinforcement learning (RL) offers a more scalable route, to the extent that it allows models to adapt and refine behaviours dynamically. However,

current LLMs, having never been exposed to structured multilingual reasoning trajectories during pre-training or SFT, cannot readily produce the reasoning patterns RL requires. A dedicated cold-start stage is therefore indispensable: it must introduce the format of parallel multilingual reasoning without undermining broader abilities. This challenge is compounded by the scarcity and high cost of multilingual reasoning traces, which explains why previous RL-based approaches have been largely confined to narrow or synthetic domains.

To address these limitations, we introduce **Polyglot-R1**, a reinforcement learning framework explicitly designed to cultivate multilingual, multi-perspective reasoning in LLMs. Our progressive curriculum begins with supervised fine-tuning on trajectories generated from straightforward multilingual prompts, establishing the basic structures of multi-path reasoning. It then transitions to reinforcement learning, where the model actively explores and generalises these behaviours in more complex multilingual and multimodal tasks. Crucially, we develop reward designs that balance accuracy with reasoning structure, providing the first empirical analysis of how multilingual reasoning evolves over training—shifting from early exploration across languages to later-stage verification across perspectives.

Our contributions are threefold:

- We propose a reinforcement learning framework that instils multilingual multi-perspective reasoning as a transferable capacity, rather than as a by-product of supervised data imitation.

- We demonstrate that Polyglot-R1 not only improves accuracy but also reshapes model strategies, showing a clear progression from exploratory reasoning to verification-oriented reasoning across languages.

- We validate multilingual reasoning as an intermediate exploration scaffold, showing how this temporary phase unlocks more robust and generalisable reasoning capabilities across linguistic boundaries.

## 2 METHODS

Existing approaches to multi-path reasoning in LLMs (Yao et al., 2023a) rely on supervised fine-tuning (SFT), which is costly, domain-limited, and encourages imitation rather than transferable reasoning challenges that are exacerbated in multilingual contexts where high-quality traces are concentrated in a few languages. To address this, we propose **Polyglot-R1**, a framework for *multilingual reasoning* that replaces expensive data pipelines with a lightweight *cold-start stage* of simpler multilingual tasks, establishing structural grounding later generalised through reinforcement learning on harder multilingual and multimodal problems. We further compare two regimes—behavioural exploration without architectural changes versus inductive biases enforcing language-independent reasoning paths—thereby disentangling the contributions of exploration and structural constraints.

In the following sections, we describe the key components of *Polyglot-R1*. First, we formulate what we mean by *multilingual multi-perspective reasoning* and detail how it is instantiated at inference time. We then present our scalable data pipeline, which generates high-quality multilingual traces for the cold-start stage. Finally, we introduce our reinforcement learning training recipes, including reward design strategies that balance accuracy and reasoning structure, and we analyse how reasoning behaviours evolve during training.

### 2.1 MULTILINGUAL MULTI-PERSPECTIVE REASONING

In human problem-solving, moments of uncertainty or ambiguity often prompt individuals to generate and compare alternative viewpoints before reaching a decision. We extend this intuition to LLMs by formalising *parallel multilingual reasoning* as a two-stage process.

**Divergence** When the model encounters a critical step, it suspends the main reasoning chain and initiates $N$ independent trajectories across languages or perspectives. Each trajectory develops autonomously, capturing linguistic and cultural diversity in problem-solving.

**Convergence** Once divergence is complete, the model aggregates the outputs, distils key insights, and reconciles conflicts into a coherent conclusion, before resuming reasoning with this synthesised representation.

This process can recur adaptively whenever required. To realise such behaviour, we employ structured control tokens: `<Parallel>` for initiating divergence, `<Path>` for independent trajectories, and `<Summary>` for aggregation. During inference, the model generates auto-regressively until predicting a `<Parallel>` token, at which point it spawns multilingual paths within `<Path>` blocks. Completion is followed by a `<Summary>` block, which integrates insights and returns to the main reasoning chain. This dynamic workflow ensures that reasoning evolves in ways that are both linguistically diverse and structurally adaptive.

## 2.2 A Scalable Data Pipeline for Multilingual Reasoning

A central challenge lies in the scarcity of high-quality multilingual reasoning traces. Although humans naturally reason in parallel, the outputs we observe are typically compressed into monolingual summaries, making such data rare in natural language corpora. Existing work has attempted to leverage the inherent parallelism of long chain-of-thought (CoT) sequences, but these methods rely on complex and computationally expensive pipelines with limited scalability.

Our approach builds on a key empirical finding: while complex multilingual problems rarely yield valid multi-perspective reasoning traces through prompting, simpler multilingual tasks consistently do. By carefully designing zero-shot prompts across multiple languages, we generate a cold-start corpus that captures the structural format of parallel reasoning. This data is not intended to solve final target tasks, but rather to familiarise the model with the structural conventions of multilingual reasoning.

To ensure quality, we implement a strict *Multilingual Reasoning Format Check*, which validates adherence to the `<Parallel>`, `<Path>`, and `<Summary>` structure across languages. This stage provides the foundation for subsequent reinforcement learning. Crucially, the design reduces dependence on large-scale annotation pipelines and enables a lightweight but practical entry point for multilingual reasoning.

## 2.3 Reinforcement Learning for Multilingual Reasoning

Once the model has acquired the structural ability to produce multilingual reasoning traces, it transitions to a reinforcement learning phase. This stage allows the model to explore and refine strategies for more complex tasks that involve both multilingual and multimodal reasoning.

We investigate two training regimes. In the *causal variant*, the architecture remains unchanged, and the model learns to balance exploration and synthesis directly through reinforcement learning. In the *structured variant*, inductive biases are introduced via modified self-attention masks and multilingual positional encodings, enforcing a degree of independence across reasoning paths in different languages.

Reward design plays a decisive role. Optimising solely for final accuracy risks the model abandoning multi-perspective reasoning in favour of shortcuts, while focusing exclusively on structural rewards leads to the overuse of scaffolds without corresponding gains in quality. To reconcile these tendencies, we adopt an alternating reward schedule: outcome-based rewards reinforce correctness, while structure-based rewards encourage the controlled use of multilingual reasoning paths.

This design provides two critical benefits. First, it prevents multi-perspective reasoning from being reduced to a superficial stylistic feature. Second, it enables us to observe how multilingual reasoning evolves strategically:

in the early stages of training, reasoning paths function primarily as exploratory mechanisms across languages, while in later stages they become verification tools, offering multi-perspective checks that strengthen the reliability of solutions.

## 2.4 A Simple and Scalable Data Pipeline for Multilingual Reasoning

Collecting high-quality data for multi-perspective reasoning is a significant challenge. Although humans naturally consider multiple possibilities in parallel, the linguistic outputs we observe are almost always compressed into a single summary. As a result, explicit traces of parallel reasoning are scarce in natural corpora. Previous approaches, such as Yang et al. (2025b), attempt to exploit the latent parallelism of long chain-of-thought sequences. Yet these methods depend on complex, multi-stage pipelines that, while avoiding costly human annotation, remain computationally intensive and limited in scalability.

Our preliminary experiments suggest a more practical route. While simple prompting fails to elicit valid reasoning traces for complex multilingual problems, it proves highly effective for more manageable tasks. Building on this insight, we design a lightweight yet scalable pipeline that uses detailed zero-shot prompts across multiple languages to generate a large corpus of well-formed reasoning traces. Crucially, this data is not intended to solve final target tasks, but rather to familiarise the model with the structural conventions of multilingual multi-perspective reasoning.

Because our structured model variant (§ 2.6) introduces architectural constraints such as path-window attention masks, strict format adherence is essential. We therefore implement a *Multilingual Reasoning Format Check*, detailed in Algorithm 1, which filters outputs to ensure consistency with the `<Parallel>`, `<Path>`, and `<Summary>` schema. This cold-start dataset provides a reliable basis from which reinforcement learning can build, allowing us to move away from data-heavy pipelines towards an approach that incrementally elicits and strengthens multilingual reasoning capabilities.

## 2.5 Eliciting Multilingual Reasoning via Reinforcement Learning in Causal Models

Unlike previous approaches, which rely on expensive data pipelines, we exploit the cold-start corpus to bootstrap reasoning structure and then extend it through reinforcement learning (RL). This stage enables the model to move from simply reproducing formatted traces to actively exploring and generalising multi-perspective reasoning on more complex multilingual tasks.

### 2.5.1 Reinforcement Learning Algorithm for Multilingual Reasoning

At the core of **Polyglot-R1** is the capacity to generate and synthesise reasoning across languages. Unlike prior work, which confines parallel paths to a single language, our framework treats each path as a distinct multilingual trajectory. A problem posed in English, for example, may be reframed in Spanish, French, or Chinese, ensuring that reasoning benefits from diverse linguistic and conceptual perspectives rather than a single lens.

We adopt Group Relative Policy Optimisation (GRPO) (Shao et al., 2024) as our reinforcement learning algorithm. Let $q$ denote a question, and $\{o_i\}_{i=1}^{G}$ the $G$ candidate responses sampled from the old policy $\pi_{\theta_{\text{old}}}(\cdot \mid q)$. Each response may be expressed in a different language or perspective. The reward $r_i$ for $o_i$ thus evaluates not only task correctness but also whether the response conforms to the multilingual reasoning structure. Formally:

$$\rho_i = \frac{\pi_\theta(o_i \mid q)}{\pi_{\theta_{\text{old}}}(o_i \mid q)}, \quad \overline{r} = \frac{1}{G}\sum_{j=1}^{G} r_j, \quad A_i = \frac{r_i - \overline{r}}{\sqrt{\frac{1}{G}\sum_{j=1}^{G}(r_j - \overline{r})^2} + \varepsilon_{\text{stab}}},$$

where $\varepsilon_{\mathrm{stab}}$ is a stability constant. The GRPO loss is then:

$$\mathcal{L}_{\mathrm{GRPO}}(\theta) = \mathbb{E}_{\substack{q \sim \mathcal{D} \\ \{o_i\} \sim \pi_{\theta_{\mathrm{old}}}}} \left[ \frac{1}{G} \sum_{i=1}^{G} \min\bigl(\rho_i A_i, \ \mathrm{clip}(\rho_i, 1-\alpha, 1+\alpha)\, A_i\bigr) \ - \ \beta\, D_{\mathrm{KL}}\bigl(\pi_\theta \,\|\, \pi_{\mathrm{ref}}\bigr) \right].$$

**Multilingual Rollout Process.** During training and inference, the model alternates between autoregressive generation, multilingual parallel exploration, and summarisation. It generates a prefix until predicting a `<Parallel>` token, at which point it launches multiple `<Path>` segments, each potentially realised in a different language. For example, one path may reason algebraically in English, another may leverage mathematical terminology from Mandarin, while a third reformulates the problem in Arabic or Spanish. Once all paths are complete, the model produces a `<Summary>` block that integrates these cross-linguistic perspectives into a coherent continuation. This iterative cycle allows reasoning to benefit from the diversity of linguistic structures and cultural framings encoded in the model's training.

This multilingual extension of parallel reasoning has two major advantages. First, it increases the likelihood of discovering complementary reasoning strategies by exploiting the diversity of linguistic framing. Second, the summarisation stage forces the model to reconcile potentially divergent linguistic insights, thereby enhancing both robustness and generalisation.

### 2.5.2 REWARD DESIGN FOR MULTILINGUAL REASONING

Designing effective rewards is central to ensuring that multilingual multi-perspective reasoning emerges as a genuine skill rather than a superficial pattern. Simply rewarding correctness ($R_{\mathrm{acc}}$) often leads models to bypass multi-path reasoning in favour of shortcuts, while rewarding structure alone encourages overproduction of parallel blocks without improving quality. In multilingual settings, there is the added challenge of encouraging *linguistic diversity* without sacrificing accuracy or coherence.

To balance these factors, we define a composite reward:

$$R_{\mathrm{final}} = R_{\mathrm{acc}} \ + \ \lambda_1 R_{\langle \mathrm{Parallel} \rangle} \ + \ \lambda_2 R_{\mathrm{div}},$$

where:

- $R_{\mathrm{acc}}$ evaluates whether the final answer is correct, independent of the languages used.
- $R_{\langle \mathrm{Parallel} \rangle}$ incentivises the use of well-formed multilingual parallel reasoning blocks (`<Parallel>`, `<Path>`, `<Summary>`).
- $R_{\mathrm{div}}$ encourages linguistic diversity across paths, rewarding models that produce reasoning in distinct languages or linguistic styles rather than duplicating content.

The diversity reward $R_{\mathrm{div}}$ is computed by measuring the language identity and lexical overlap between paths. Positive reward is assigned when the model employs at least two distinct languages or registers within a parallel block. At the same time, penalties are applied when all paths collapse into near-identical reasoning.

We further adopt an *alternating schedule*, where training alternates between episodes focused on accuracy and those focused on structure and diversity. This prevents the model from overfitting to a single objective and encourages it to learn when multilingual reasoning is genuinely beneficial. In early training, diversity is primarily used as an exploration mechanism: different languages introduce alternative framings that expand the solution space. Later, multilingual reasoning acts as a verification strategy, where cross-lingual perspectives are compared and consolidated to strengthen confidence in the final answer.

This reward design anchors multilingual multi-perspective reasoning as more than a formatting trick: it becomes an adaptive skill that both broadens exploration and sharpens verification.

### 2.5.3 TRAINING RECIPE AND REWARD DESIGN

The training process unfolds in three stages:

**Cold-Start Stage.** Using the corpus described in Section 2.4, we fine-tune the initial actor on a small set of multilingual reasoning traces. This stage is not designed to teach solutions but to ensure the model can reliably produce structured reasoning formats.

**RL on Simple Multilingual Tasks.** After cold start, the model can generate reasoning tags but the behaviour is unstable, as these tokens never appeared in pre-training. We therefore perform small-scale RL to consolidate format learning. Here, the final reward is $R_{final} = R_{\langle\text{Parallel}\rangle} \times R_{\text{acc}}$, combining correctness ($R_{\text{acc}}$) and structural adherence ($R_{\langle\text{Parallel}\rangle}$). A reward of +1 is given only if at least one valid parallel reasoning unit is produced *and* the final answer is correct; otherwise, the model receives -1.

**RL on General Multilingual Tasks.** Having stabilised format generation, the model is trained on more challenging multilingual and multimodal problems. Here, we prioritise task performance, using accuracy-based rewards only. This ensures that the model employs multi-perspective reasoning strategically rather than indiscriminately. The models obtained at this stage constitute our *Polyglot-Seen* variants.

## 2.6 ELICITING MULTILINGUAL REASONING VIA RL

While the causal variant learns reasoning behaviours directly, hidden representations may leak across reasoning paths, undermining independence. To counter this, we propose a structured variant, *Polyglot-Unseen*, which embeds inductive biases into the architecture. Specifically, we use *path-window masking*, which restricts tokens in a `<Path>` block to attend only to their own context and the shared prefix, and *multiverse positional encodings*, which assign disjoint position indices to each path. Together, these mechanisms enforce independence among reasoning threads while preserving visibility from the `<Summary>` block, where cross-lingual synthesis occurs.

### 2.6.1 REWARD SCHEDULES

The progressive recipe used for the causal variant proves ineffective for the structured model, as attention masks trained on simple tasks do not generalise to harder problems. To address this, we experiment with two alternative reward schedules.

The first **(S1: Accuracy-only)** optimises exclusively for correctness, providing no explicit incentive to employ parallel reasoning. The second **(S2: Alternating accuracy and structure)** alternates every $W = 10$ steps between accuracy-based and structure-aware rewards. The latter offers graded incentives: +1.2 when a valid parallel reasoning unit is produced and the answer is correct, +1.0 when no parallel unit is used but the answer is correct, and –1.0 otherwise. This alternating schedule encourages the model to apply multilingual reasoning selectively and effectively, without overfitting to superficial structural cues.

## 3 EXPERIMENTS

### 3.1 EXPERIMENTAL SETUPS

**Models.** We adopt Qwen-3-4B-Base (Yang et al., 2025a) and Llama-3-3B as backbone models. The former represents a state-of-the-art open-source LLM at the 4B scale, striking an effective balance between efficiency and performance, and is therefore well-suited for multilingual reinforcement learning experiments.

| Method | # Parallel | MAIME25 | | MGSM-Symbolic | | MSVAMP | |
|---|---|---|---|---|---|---|---|
| | | Mean@16 | Pass@16 | Mean@16 | Pass@16 | Mean@16 | Pass@16 |
| Qwen3-4B-Base | 0.0 | 1.6 | 11.4 | 3.2 | 15.9 | 7.5 | 49.8 |
| *SFT + Multilingual Parallel Reasoning* | | | | | | | |
| Polyglot-SFT-Seen | 95.6 | 8.3 | 28.6 | 11.2 | 27.1 | 47.5 | 78.4 |
| Polyglot-SFT-Unseen | 95.6 | 5.0 | 21.4 | 9.1 | 25.9 | 42.8 | 79.6 |
| *RL Approach* | | | | | | | |
| GRPO (Multilingual DAPO) | 0.0 | 15.2 | 33.1 | 17.9 | 29.8 | 62.4 | 84.6 |
| Polyglot-R1-Seen | 27.3 | **19.6** | 39.4 | **19.1** | **36.2** | **69.9** | 84.1 |
| Polyglot-R1-Unseen (S1) | 13.6 | 17.2 | 36.9 | 18.0 | 32.7 | 68.5 | 87.6 |
| Polyglot-R1-Unseen (S2) | **63.0** | 18.8 | **41.6** | 16.9 | 30.9 | 66.7 | **90.2** |

Table 1: Performance comparison on multilingual reasoning benchmarks for Qwen-3-4B-Base under different multilingual parallel reasoning configurations (Llama-3-3B in appendix). We report Mean@16 and Pass@16.

**Evaluation.** Our evaluation covers a suite of multilingual and mathematical reasoning benchmarks, including Multilingual AIME25 (MAIME25) Bianchi (2025), MGSM-Symbolic Ranaldi & Pucci (2025), and MSVAMP (Chen et al., 2024). To assess multilingual generalisation, we extend these with parallel prompts spanning both high- and low-resource languages. For each task, we generate 16 independent responses per question at a fixed temperature and report the average accuracy ($\mathrm{mean@16}$), following prior work (Wang et al., 2025b). We additionally report $\mathrm{pass@16}$ as an upper-bound measure of performance.

**Training Details.** Our implementation builds on VERL (Sheng et al., 2024), adhering closely to its training recipe with only minimal adjustments to hyperparameters. In the cold-start stage, we apply SFT on our curated POLYGLOT-MGSM dataset, using a batch size of 128, a learning rate of $1 \times 10^{-5}$, a weight decay of 0.01, and a warm-up ratio of 0.1 with a cosine learning-rate schedule. This yields 58 and 230 gradient steps for the Polyglot-SFT-Seen and Polyglot-SFT-Unseen variants, respectively.

In Stage 1, we conduct RL on multilingual variants of MGSM for five epochs, with a batch size of 1024, five rollouts, and a learning rate of $1 \times 10^{-6}$, resulting in 35 gradient steps. In Stage 2, we train on a multilingualized version of the DAPO dataset for 300 steps, using a batch size of 512, eight rollouts, and the same learning rate. No additional warm-up or scheduling is applied.

### 3.2 MAIN RESULTS

Table 1 presents the results, comparing our approach against two strong baselines: GRPO applied directly to the multilingual DAPO training set, and GRPO in two stages—first on multilingual MGSM, then on multilingual DAPO. The **Polyglot-R1** framework consistently outperforms both. The strongest performance is achieved by the causal variant (`Polyglot-R1-Seen`), whose success derives from the curriculum: the cold-start stage establishes the format of multilingual reasoning, while reinforcement learning progressively consolidates this ability. Naïve SFT delivers notable improvements over the base model, yet still falls well short of the RL baselines. We further observe a trade-off between causal and structured variants: the causal model (Seen) is more stable and yields the highest overall performance, whereas structured models (Unseen) are more sensitive to reward design. Among these, the alternating reward schedule (S2) offers the best compromise, combining high levels of parallel reasoning with strong accuracy.

### 3.3 ABLATION

Removing Stage 1 RL on MGSM leads to a consistent performance drop (average 2.3%) in the causal variant, showing that SFT alone cannot reliably activate or sustain parallel reasoning. In contrast, Stage 1 RL reduces

| Training Configuration | MAIME25 | MGSM-Symbolic | MSVAMP |
|---|---|---|---|
| **Effect of Training Stages** | | | |
| Polyglot-R1-Seen | **19.6** | **19.1** | **69.9** |
| - w/o RL on Multilingual GSM8K | 18.1 | 18.7 | 65.4 |
| Polyglot-R1-Unseen (S1) | 17.4 | 18.0 | 68.7 |
| + with RL on Multilingual GSM8K | 14.7 | 13.4 | 53.2 |
| **Effect of Multilingual Prompting** | | | |
| Polyglot-R1-Seen | **19.6** | **19.1** | **69.9** |
| - w/o Multilingual Prompt | 20.2 | 16.7 | 66.1 |

Table 2: Ablation study on training approach: comparison of different multilingual training configurations.

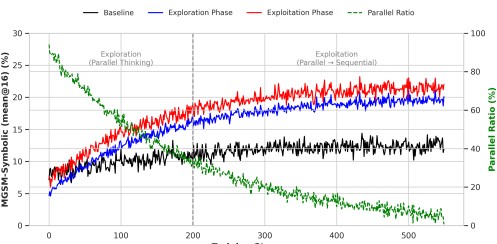

Figure 1: Two-stage training with parallel reasoning as a mid-training exploration scaffold. Stage 1 promotes exploration through alternating rewards, while Stage 2 consolidates accuracy via sequential reasoning. Accuracy continues to improve even as explicit parallel reasoning becomes less frequent.

performance in the structured variant (average 8.6%), as attention masks learned on simple multilingual tasks transfer poorly to harder ones. This suggests that causal models require Stage 1 RL to bootstrap multilingual reasoning, while structured models demand different training strategies. Eliminating explicit multilingual prompts (Table 2) lowers performance by up to 1.8%, indicating that such prompts help the model internalise multilingual reasoning rather than merely replicate monolingual patterns.

Finally, reward design proves critical. Accuracy-only optimisation improves correctness but yields very low parallel usage, while structure-only maximises the parallel ratio but sacrifices accuracy by overusing reasoning blocks without substantive gains. Alternating accuracy and structure provides the best trade-off, maintaining accuracy while encouraging effective multilingual reasoning and outperforming accuracy-only baselines on more challenging tasks.

### 3.4 EVOLUTION OF MULTILINGUAL REASONING BEHAVIOUR

We monitored the relative position of the `<Parallel>` block throughout training (Figure 4). In the early stages, multilingual reasoning tends to appear near the beginning of the solution, reflecting its exploratory use for broadening the search space. As training advances, the `<Parallel>` blocks progressively shift towards later stages of the reasoning chain, where they function as a verification mechanism: alternative linguistic framings are compared to confirm the solution.

This dynamic illustrates a clear strategic transition. Initially, linguistic diversity drives exploration; later, it enhances reliability and truthfulness by consolidating consistency across languages.

## 3.5 PARALLEL MULTILINGUAL REASONING AS A MID-TRAINING SCAFFOLD

A central challenge in reinforcement learning is maintaining sufficient exploration. Enforcing multilingual reasoning during mid-training biases the model towards a broader policy search. In Stage 1, alternating rewards sustain linguistic diversity; in Stage 2, the focus shifts to accuracy, allowing the model to consolidate the most effective strategies. Empirically, this scaffold raises the performance ceiling: accuracy improves even as explicit multilingual reasoning blocks decline (Figure 1), showing that multilingual reasoning serves both as a direct aid to problem-solving and as a structured exploration mechanism.

## 4 RELATED WORK

**Parallel Thinking**    Parallel thinking has recently become a central topic in reasoning research (Yao et al., 2023b; Wang et al., 2022; Brown et al., 2024; Pan et al., 2025; Huang et al., 2024; Hsu et al., 2025; Rodionov et al., 2025; Yang et al., 2025b; Zheng et al., 2025). Early work often relied on brute-force strategies, spawning trajectories at the outset and merging them at the end (Brown et al., 2024; Wang et al., 2022), or synchronising partial solutions at fixed intervals (Rodionov et al., 2025; Hsu et al., 2025). These approaches are limited, as branching and aggregation follow pre-defined schedules rather than the dynamics of reasoning itself. Structured methods such as Monte Carlo Tree Search (Zhang et al., 2024) and Tree of Thoughts (Yao et al., 2023b) offer finer control but remain dependent on heuristics and external verifiers. More recent work explores reinforcement learning for adaptive parallelism, although it is often restricted to efficiency gains or monolingual toy tasks (Zheng et al., 2025). By contrast, we argue that reinforcement learning provides a more general route: it maintains efficiency while enabling adaptive behaviours. Our contribution extends this to the multilingual setting, where parallel thinking must contend with linguistic diversity, an essential step for equitable model deployment across languages.

**Improving Reasoning via Reinforcement Learning with Verifiable Rewards**    RLVR optimises models using automatically checkable outcome-based rewards, removing the need for reward models or detailed human annotation. It has proven effective across domains including mathematics (Guo et al., 2025), code (Wang et al., 2025a), multimodal reasoning (Huang et al., 2025b; Li et al., 2025), relation extraction (Dai et al., 2025), and GUI navigation (Shi et al., 2025). Work on efficiency and stability has introduced innovations such as self-play (Liu et al., 2025; Huang et al., 2025a), test-time RL (Zuo et al., 2025), and robust algorithms like DAPO (Yu et al., 2025), VAPO (Yue et al., 2025), and entropy-guided optimisation (Wang et al., 2025b). Yet significant challenges persist. Faithfulness (Chen et al., 2025; Zhou et al., 2025) and robustness (Sabbaghi et al., 2025) remain unresolved, and most work assumes strictly sequential reasoning. This is a fundamental limitation, since LLMs do not naturally perform parallel rationale, and current RLVR methods cannot instil it. We address this gap by introducing a progressive curriculum that enforces and then consolidates multilingual parallel reasoning, equipping models with adaptive, multi-perspective reasoning capabilities across languages.

## 5 CONCLUSION

We introduced Polyglot-R1, the first reinforcement learning framework designed to cultivate multilingual, multi-perspective reasoning. Its progressive curriculum combines a lightweight cold-start stage with reinforcement learning and carefully structured rewards, yielding consistent gains in accuracy on demanding reasoning benchmarks. Our analysis reveals that multilingual reasoning develops strategically: initially serving as a source of cross-linguistic diversity and later functioning as a mechanism for verification across perspectives. Crucially, we establish multilingual reasoning as a mid-training scaffold, demonstrating that enforced diversity during intermediate phases of training enables more robust and transferable final performance.

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

# A PROMPTS

**Baseline Prompt**

{Problem} Let's think step by step and output the final answer after "Final Answer: ".

**Multilingual Thinking Prompt**

Solve the following problem step by step. During the reasoning process, whenever you encounter a step that may benefit from cross-linguistic perspectives, insert a <Parallel> block at that point.

Within each <Parallel> block:
- Include at least two distinct reasoning paths, each framed in different languages.        - Each path must be enclosed within <Path lang="..."> and </Path> tags.               - Do not include ordering or cross-references between paths, as they are generated independently. - Close the block with </Parallel>.

Immediately after each </Parallel>, provide a concise summary that integrates insights across languages, enclosed in <Summary> and </Summary> tags.

Repeat this process adaptively throughout the reasoning chain. Do not explicitly mention that you are using multilingual reasoning|just insert the <Parallel> block naturally when helpful.

End your response with a line starting with Final Answer: followed by the final result.

Problem: {Problem}

## B    ABLATION STUDY

| Training Configuration | Parallel Ratio | MAIME25 | MGSM-Symbolic | MSVAMP |
|---|---|---|---|---|
| Accuracy | 13.6 | 17.7 | **18.3** | 69.2 |
| Parallel | **80.3** | 17.4 | 15.0 | 59.9 |
| Alternating Acc./Parallel | 63.0 | **18.9** | 16.2 | **67.8** |

Table 3: Ablation study on reward modelling for the POLYGLOT-R1-UNSEEN model.

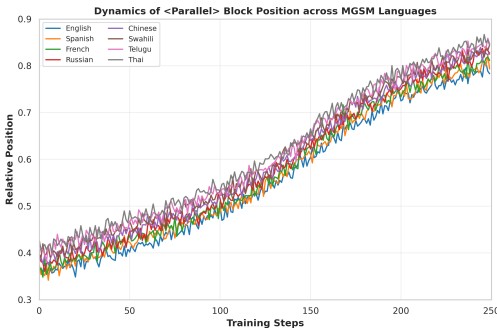

Table 4: Dynamics of the relative position of the `<Parallel>` block during RL training. The upward trend indicates a gradual shift from exploratory use of parallel reasoning towards later verification stages.

## C    ADDITIONAL RESULTS

| Method | # Parallel | MAIME25 | | MGSM-Symbolic | | MSVAMP | |
|---|---|---|---|---|---|---|---|
| | | Mean@16 | Pass@16 | Mean@16 | Pass@16 | Mean@16 | Pass@16 |
| Llama-3-3B-Base | 0.0 | 1.3 | 10.2 | 3.8 | 16.7 | 8.2 | 48.5 |
| *SFT + Multilingual Parallel Reasoning* | | | | | | | |
| Polyglot-SFT-Seen | 95.6 | 7.5 | 26.9 | 11.8 | 28.4 | 45.9 | 76.8 |
| Polyglot-SFT-Unseen | 95.6 | 4.6 | 19.7 | 9.4 | 24.6 | 41.2 | 78.1 |
| *RL Approach* | | | | | | | |
| GRPO (Multilingual DAPO) | 0.0 | 14.4 | 31.8 | 18.5 | 30.5 | 61.1 | 83.2 |
| Polyglot-R1-Seen | 27.3 | **18.9** | 38.2 | **19.7** | **35.6** | **68.4** | 82.7 |
| Polyglot-R1-Unseen (S1) | 13.6 | 16.8 | 35.1 | 18.3 | 31.9 | 67.1 | 86.0 |
| Polyglot-R1-Unseen (S2) | **63.0** | 18.1 | **40.2** | 17.2 | 30.1 | 65.5 | **88.9** |

Table 5: Performance comparison on multilingual and mathematical reasoning benchmarks for Llama-3-3B under different multilingual parallel reasoning configurations (results for Qwen-3-4B in main text). We report Mean@16 and Pass@16.

---

**Algorithm 1** Multilingual Parallel Thinking Format Check

---

**Require:** tokens: list of tokens from the multilingual parallel-thinking trace; tag-pairs: set of valid (opening, closing) tag
    pairs, e.g. $\{(\texttt{<Path lang="en">},\texttt{</Path>}),(\texttt{<Path lang="es">},\texttt{</Path>}),\dots\}$
**Ensure:** *format_valid*: boolean indicating whether the trace is well-formed
1:  $S \leftarrow \emptyset$
2:  *format_valid* $\leftarrow$ true
3:  **for** each $t \in$ tokens **do**
4:      **if** $t$ is an opening tag (with valid language attribute) **then**
5:          push $t$ onto $S$
6:      **else if** $t$ is a closing tag **then**
7:          **if** $S$ is empty **then**
8:              *format_valid* $\leftarrow$ false
9:              **break**
10:         **end if**
11:         $top\_tag \leftarrow \text{Top}(S)$
12:         **if** $(top\_tag, t) \in$ tag-pairs **then**
13:             pop $S$
14:         **else**
15:             *format_valid* $\leftarrow$ false
16:             **break**
17:         **end if**
18:     **end if**
19: **end for**
20: **if** *format_valid* **and** $S \neq \emptyset$ **then**
21:     *format_valid* $\leftarrow$ false
22: **end if**
23: **return** *format_valid* =0

---

