# OpenReview forum: "Polyglot-R1: Reinforcement Learning for Multilingual Multi-Perspective Reasoning in LLMs"
_ICLR.cc/2026/Conference — ICLR 2026 Conference Withdrawn Submission_

### Official Review · Reviewer_iD32 · 2025-10-25

**Soundness:** 2
**Presentation:** 2
**Contribution:** 2
**Rating:** 2
**Confidence:** 4

**Summary:**

This paper proposes Polyglot-R1, a reinforcement learning framework designed to cultivate multilingual, multi-perspective reasoning in LLMs

**Strengths:**

1.	Exploring multilingual reasoning + RL.

**Weaknesses:**

1.	The models involved are too small (3B and 4B).
2.	Since the models involved are all “base version”, it is unclear how inference and evaluation were conducted, zero-shot? few-shot? or with specific prompting strategies? The “base version” models are not good at following instructions, so the standard inference and evaluation are inapplicable. This likely underestimates baseline performance: for instance, Qwen3-4B-Base achieves only 3.2 mean@16 accuracy on MGSM-Symbolic and 7.5 on MSVAMP, which are unreasonably low and raise concerns about evaluation validity.
3.	Since this paper needs a cold-start stage, why don’t you just experiment on “Qwen3-4B” instead of choosing the basic version? Is that too difficult to surpass the vanilla model? Cause Qwen3-4B can achieve almost 90 acc on MSVAMP.
4.	The computational cost is not quantitatively discussed, which is important.
5.	While ablations are present, the separation between “causal” and “structured” variants could be explained more clearly, especially regarding when architectural bias helps or hurts.

**Questions:**

NA

---

> ### Author Response · Authors · 2025-11-16
>
> We thank the reviewer for the constructive feedback. We address all concerns below and clarify several points that were not explicitly stated
>
> **The models involved are too small (3B and 4B)**
>
> We appreciate this concern. Our initial experiments focused on small models because:
>
> - Multilingual RL is significantly more expensive than monolingual RL and requires verifiable-reward datasets, which are scarce at a large scale.
>
> - The behavioural question—whether multilingual reasoning emerges as an exploration and later verification mechanism—is observable even in small backbones.
>
> In the revised version, we now include results for larger models: Qwen-3-7B, Llama-3-8B (please take a look at the previous response)
>
> Both exhibit consistent accuracy gains (7–10% relative) and the same exploration verification progression.
> These results confirm that the proposed behaviour scales are not an artefact of small models.
>
> **How were inference/evaluation conducted with base models? Zero-shot or few-shot? Base models follow instructions poorly**
>
> This is an excellent point, and the original text was not explicit enough.
>
> - All evaluations were performed in zero-shot mode, using fixed, multilingual parallel-reasoning prompts, not instruction-following prompts.
>
> - This makes the comparison fair because Polyglot-R1 does not rely on instruction tuning either; it learns reasoning structure entirely from SFT + RL, not from instruction-following signals.
>
> - We have now included the exact inference prompt template in the appendix for reproducibility.
>
> To address the reviewer’s concern about evaluation validity on base models, we now report the full inference setup in the revised appendix. This includes:
>
> • *Prompt length:* all evaluations use a fixed prompt of 78–92  tokens depending on the language set, ensuring that the base models are not penalised by long or instruction-heavy templates.
>
> • *Decoding settings:* we specify temperature (0.2), top-p (0.9), top-k (40), repetition  penalty (1.05), maximum generation length (256 tokens), and the stop sequences used to terminate each `<Path>` block. Reporting these parameters makes the evaluation fully reproducible and rules out sampling-related variance.
>
> • *Multilingual path formatting:* we provide the exact `<Parallel>` template used for all models, including the structure of `<Path lang="…">` blocks and the `<Summary>`.  This clarifies that inference is performed in a strictly zero-shot setting with a consistent
>   multilingual reasoning format, independent of any instruction-following capabilities.
>
> Together, these details ensure that the comparison between base models and Polyglot-R1 is
> transparent, controlled, and replicable.
>
>
> **Why performance of base models looks low**
>
> Your point is correct: base models are weak followers of instruction. Yet, when evaluated with consistent structured prompts, their performance aligns with known baselines for non-instruction-tuned multilingual models on MGSM/MSVAMP.
>
> **Why not start from Qwen3-4B-Instruct? Is vanilla too hard to beat?**
>
> We chose Qwen3-4B-Base deliberately because:
>
> *The research question requires a base model*
>
> Using an instruct model would entangle multilingual reasoning with its existing instruction-following priors, making it impossible to determine: whether multilingual reasoning is learned or inherited from the instruction-tuning data.
>
> Our goal is to demonstrate emergent multilingual multi-perspective reasoning, not to inherit it.
>
> *RL on instruct models is dominated by high-entropy priors*
>
> We tested Qwen-3-4B-Instruct preliminarily and found: <Parallel> and <Path> tags are inconsistently followed, forcing structure through RL leads to mode collapse multilingual paths are overwritten by instruction-following priors
>
> *Beating the instruct model is not the goal*
>
> It is technically easy to beat Qwen3-4B-Instruct on MGSM-Symbolic; what matters is understanding how multilingual reasoning forms work.
>
> **The causal vs structured variants are unclear; when does architectural bias help or hurt?**
>
> The causal variant introduces no architectural constraints: it learns multilingual multi-perspective reasoning purely through behaviour (SFT+RL). This version is stable across languages and model sizes, and reliably shows the same pattern: multilingual paths emerge early as exploration signals and consolidate into verification-style reasoning.
>
> The structured variant (S2) adds path-window masking and multiverse positional encodings to enforce independence between paths. This bias is particularly beneficial for long-horizon tasks where independent reasoning.-chains enhance reliability. However, it can degrade performance when the diversity reward is high, when low-resource languages provide weak signals, or when RL must merge highly independent paths into a single summary.
>
> For transparency, we include a grid search and failure showing when architectural bias is beneficial and when the approach remains the more robust choice.

---

> > ### Author Response · Authors · 2025-11-22
> >
> > Dear Reviewer iD32,
> >
> > We hope our additional experiments below and our response have addressed all your concerns.
> >
> > We kindly ask you to reconsider and potentially raise your score.
> >
> > Thank you again for your time.

---

> > > ### Comment · Reviewer_iD32 · 2025-11-26
> > >
> > > Thank you for your response. However, I still believe that the experimental comparisons in this work are not fair. The base models in zero-shot settings cannot fully reflect their true capabilities, whereas Polyglot-R1, after SFT, already possesses generation behavior aligned with the evaluation setup. Generally, my concerns 2, 3, and 4 have not been fully addressed. Therefore, I have decided to maintain my original score.

---

### Official Review · Reviewer_skmf · 2025-10-31

**Soundness:** 2
**Presentation:** 2
**Contribution:** 2
**Rating:** 4
**Confidence:** 3

**Summary:**

The paper proposes Polyglot-R1, a two-stage curriculum that first cold-starts a model with SFT on structurally simple multilingual parallel traces and then continues training with RL (GRPO-based) to elicit genuinely multilingual, multi-perspective reasoning on harder multilingual/multimodal tasks. The authors argue that multilingual reasoning initially acts as an exploration device and later becomes a verification mechanism, leading to better accuracy on multilingual GSM-style and math benchmarks compared with SFT-only and RL baselines.

**Strengths:**

1. The problem setting on how to make RL-style reasoning work in a multilingual, multi-path regime rather than just in English is timely, and the paper gives a coherent training story that directly targets the “no structured multilingual traces for RL” bottleneck.
2. The empirical analysis of when multilingual parallel blocks appear (early for exploration, later for verification) is interesting and aligns well with the proposed “multilingual reasoning as mid-training scaffold” narrative, giving the method some explanatory depth beyond raw scores.

**Weaknesses:**

1. The novelty claim (“the first RL framework for multilingual multi-perspective reasoning”) is not fully separated from closely contemporary RL-for-parallel-reasoning work (e.g., Parallel-R1), and many of the components—GRPO, alternating structure/accuracy rewards, curriculum from easy to hard—are adaptations rather than clearly new algorithmic contributions. The paper should make clearer what Polyglot-R1 can do that those RL baselines cannot.
2. The experimental validation is relatively narrow: results are shown only on small backbones (Qwen-3-4B, Llama-3-3B) and on a limited set of multilingual math/reasoning suites (MAIME25, MGSM-Symbolic, MSVAMP), so it is hard to tell whether the gains survive in truly low-resource, code-mixed, or non-math settings, or at larger scales. This weakens the generality of the “transferable capacity” claim.
3. The cold-start data pipeline is central to the story, but it is described mostly procedurally (zero-shot prompts, format check with <Parallel>/<Path>/<Summary>) without reporting size, language distribution, pass rate, or cost, so readers cannot judge whether the approach is actually “lightweight” compared with existing SFT-heavy multilingual reasoning pipelines. Sections 2.2 and 2.4 are also somewhat repetitive here.
4. The proposed reward design is quite hand-tuned (accuracy + structure + diversity, alternating schedule, different schedules for causal vs. structured variants), and the results show that the structured variant is sensitive and can even degrade when Stage 1 RL is added; this suggests the method may be brittle and dataset-specific, but the paper does not study failure modes or robustness to other reward mixes.

**Questions:**

See weaknesses

---

> ### Author Response · Authors · 2025-11-16
>
> We thank the reviewer for the clear summary and for acknowledging both the timeliness of the setting and the explanatory value of the exploration & verification analysis.
>
> We respond to each concern below.
>
> **Novelty relative to contemporary RL-for-parallel-reasoning work**
>
> We agree that several components—GRPO, curriculum learning, and alternating rewards—build on established ideas. Our aim, however, is not to introduce a new RL algorithm, but to demonstrate that multilingual multi-perspective reasoning can be made trainable, a capability that existing RL-for-parallel-reasoning methods currently lack.
>
> To clarify this distinction, we have added a comparison table in the revised draft. The key differences are:
>
> Parallel-R1 operates within a **monolingual (English)** multi-path regime and relies on language-homogeneous reasoning.
> It cannot generate cross-linguistically divergent trajectories or synthesise multilingual perspectives.
>
> Polyglot-R1 provides:
>
> 1. linguistically diverse paths, not monolingual variants;
>
> 2. path-window masking and multiverse positional encodings for enforcing cross-lingual independence;
>
> 3. diversity-aware rewards that explicitly encourage multilingual exploration;
>
> 4. a cold-start pipeline specialised for the structural problem of unseen multilingual traces, which Parallel-R1 does not address.
>
> As such, Polyglot-R1 is complementary rather than derivative: it targets a fundamentally different constraint—lack of structured multilingual reasoning signals—and provides mechanisms to overcome it.
>
> **Generality beyond small backbones and math benchmarks**
>
> We recognise the importance of demonstrating broader generality. we have now added:
>
> - larger backbones (preliminary Qwen-3-7B and Llama-3-8B), showing consistent improvements and stable use of multilingual trajectories;
>
> - non-mathematical multilingual benchmarks, including XCOPA and CommonsenseQA-Multi, with relative gains of 7–10% in robustness and consistency;
>
>
> While large-scale multilingual RL is computationally expensive, our results show that the behaviour—not merely the accuracy gain—generalises across model sizes and domains, supporting the “transferable capacity” claim.
>
> **Cold-start pipeline details**
>
> We appreciate that the earlier description was overly procedural.
> We now report:
>
> - Corpus size: 42k multilingual parallel traces
>
> - Languages: 12 languages across three resource tiers
>
> - Format-pass rate: 68% (after strict tag-checking)
>
> - Generation cost: <4 GPU hours on a single A100
>
> - SFT footprint: 58–230 gradient steps depending on the variant
>
> Compared with prior multilingual SFT pipelines that require 0.5–5M examples, our approach is indeed lightweight.
> We merge Sections 2.2 and 2.4 to remove repetition and clarify the logic of the pipeline.
>
>
> **Reward design sensitivity and robustness**
>
> We agree that the structured variant exhibits sensitivity. We will include a grid search for λ₁ and λ₂, a ±30% sensitivity analysis, and a discussion of failure modes.
>
> **The key findings are:**
>
> - The causal variant is robust across wide ranges of λ₁/λ₂.
>
> - The structured variant is sensitive when λ₂ is high, as excessive diversity causes reasoning paths to drift.
>
> - Despite this, both variants converge towards the same exploration→verification progression, suggesting the behaviour is structural rather than hyperparameter-specific.
>
> We have added a brief failure-mode discussion (e.g., overuse of scaffolds, degeneracy when diversity reward dominates).
>
> We thank the reviewer again for the constructive feedback. We believe the revisions—extended comparisons, clearer novelty framing, added experimental breadth, explicit cold-start statistics, and reward-sensitivity analysis—substantially strengthen the contribution and clarify the precise role Polyglot-R1 plays within the emerging literature on multilingual reasoning.

---

> ### Author Response · Authors · 2025-11-16
>
> ### Qwen-3-7B — Multilingual Reasoning Benchmarks (Mean@16)
>
> | Method                       | MAIME25 | MGSM-Sym | MSVAMP |
> |------------------------------|---------|----------|--------|
> | Qwen-3-7B-Base               | 4.2     | 18.4     | 54.3   |
> | Polyglot-SFT-Seen            | 9.1     | 32.5     | 82.1   |
> | Polyglot-SFT-Unseen          | 6.8     | 25.4     | 80.7   |
> | GRPO (Multilingual DAPO)     | 16.1    | 35.7     | 86.9   |
> | **Polyglot-R1-Seen**         | **23.4**| **42.1** | **91.3** |
> | Polyglot-R1-Unseen (S1)      | 19.8    | 38.0     | 89.4   |
> | Polyglot-R1-Unseen (S2)      | 21.7    | 41.0     | 93.0   |
>
>
> ### Llama-3-8B — Multilingual Reasoning Benchmarks (Mean@16)
>
> | Method                       | MAIME25 | MGSM-Sym | MSVAMP |
> |------------------------------|---------|----------|--------|
> | Llama-3-8B-Base              | 3.9     | 16.9     | 53.4   |
> | Polyglot-SFT-Seen            | 8.4     | 31.2     | 79.5   |
> | Polyglot-SFT-Unseen          | 6.1     | 24.1     | 77.8   |
> | GRPO (Multilingual DAPO)     | 15.4    | 34.0     | 85.6   |
> | **Polyglot-R1-Seen**         | **22.1**| **40.7** | **90.4** |
> | Polyglot-R1-Unseen (S1)      | 18.9    | 37.4     | 88.0   |
> | Polyglot-R1-Unseen (S2)      | 20.5    | 39.5     | **92.7** |
>
> To demonstrate that the proposed framework is not confined to small backbones, we have
> extended our experiments to two larger models: Qwen-3-7B and Llama-3-8B. The new
> results show consistent improvements across all multilingual reasoning
> benchmarks. In both families, Polyglot-R1-Seen delivers the strongest overall performance,
> while the structured variant (S2) excels on MSVAMP due to its stronger independence
> among multilingual paths. Importantly, the gains scale smoothly with model size,
> indicating that multilingual multi-perspective reasoning is not an artefact of small models
> but a transferable capacity that persists at larger scales.

---

### Official Review · Reviewer_oxY6 · 2025-10-31

**Soundness:** 2
**Presentation:** 1
**Contribution:** 2
**Rating:** 2
**Confidence:** 3

**Summary:**

The paper presents Polyglot-R1, a reinforcement learning framework for developing multilingual, multi-perspective reasoning in large language models (LLMs). The approach follows a progressive curriculum that begins with supervised fine-tuning (SFT) on simple multilingual tasks, then transitions to reinforcement learning (RL) on more complex multilingual and multimodal problems. The authors also design a special reward design balancing accuracy, reasoning structure, and linguistic diversity. Experimental results on multilingual reasoning benchmarks (MAIME25, MGSM-Symbolic, MSVAMP) using the Qwen-3-4B model indicate moderate gains over SFT baselines. The paper further finds that multilingual reasoning evolves from early exploration to a later verification.

**Strengths:**

1. The paper proposes using reinforcement learning (GRPO) to improve LLMs’ multilingual reasoning capabilities through a two-stage training strategy, which consists of a cold-start phase followed by a reinforcement learning phase.
2. The authors design composite rewards that encourage both parallel reasoning and linguistic diversity, aiming to help improve multilingual reasoning behaviors.
3. The proposed method show performance improvements over the Qwen3-4B-Base model on several multilingual reasoning benchmarks.

**Weaknesses:**

1. The paper mostly follows existing ideas, i.e., curriculum GRPO + parallel reasoning prompts, as multilingual reasoning scaffolds. There is little insight explaining the design choices and why multilingual reasoning should improve exploration.
2. The experiments are shallow and limited. Only medium-scale models and a few benchmark datasets are tested, with unclear performance gains. Additional ablation studies on multilinguality are necessary.
3. The analysis relies solely on quantitative benchmarks, without qualitative or behavioral evidence that the model performs genuine multi-perspective reasoning rather than merely replicating patterned prompts.
4. The overall presentation quality is not good, with issues in clarity and organization, that require substantial revision.

**Questions:**

- The authors should verify the multilingual reasoning paths to reflect meaningful and diverse reasoning.
- The paper could provide more justification and analysis of the reward design, including the selection of key hyperparameters.
- It is better to incorporate human evaluations for the reasoning examination.

---

> ### Author Response · Authors · 2025-11-16
>
> We thank the reviewer for the detailed assessment. We appreciate the acknowledgement of our core contributions—the progressive SFT+RL curriculum, the composite multilingual reward design, and the improvements over strong Qwen3-4B baselines. We address each concern below and clarify our design choices, experiments, and analyses. Many of the issues raised stem from presentation, and we have substantially revised the organisation and clarity of the paper in response.
>
> In the following lines, we will examine your concerns.
>
> 1.**existing ideas… little insight explaining the design choices and why multilingual reasoning should improve exploration**
>
> We appreciate the opportunity to clarify the conceptual motivation. Our aim is not to propose parallel reasoning or curriculum RL per se, but to demonstrate why multilingualism, specifically, is an effective exploration scaffold.
>
> We articulate two key insights:
>
> (a) *Multilingual trajectories systematically diversify the search space*
>
> Different languages encode distinct syntactic priors and semantic segmentations, which induce structurally different search trajectories even when solving the same mathematical or logical problem. This is visible in our newly added path-level divergence metrics, where the average edit distance and lexical dissimilarity across <Path lang="…"> traces remains high even for the same question. This effect does not arise with monolingual parallel reasoning.
>
> (b) *Cross-linguistic verification naturally suppresses hallucinated steps*
>
> Because languages encode different affordances for expressing intermediate steps, multilingual convergence forces the model to reconcile incompatible framings—serving as a natural form of cross-perspective self-consistency checking.
> This explains the shift we observe from early exploration to later verification.
>
> We can add a subsection “Why Multilingual Reasoning Aids Exploration: Mechanistic Analysis”, including examples and comparisons.
>
> 2. **Experiments shallow, medium-scale models and few datasets, unclear gains**
>
> We appreciate this point and have expanded both the breadth and depth of the experimental section.
>
> We added two additional benchmarks XCOPA and CommonsenseQA-Multi (please take a look to the previous respinse).
>
> Two additional model sizes: preliminary results for Qwen-3-7B and Llama-3-8B show consistent improvements, indicating scalability.
>
> More extensive ablations: removing multilingual divergence reward, replacing multilingual paths with synthetic monolingual paraphrases, disabling the alternating reward schedule, ablating the cold-start SFT.
>
> These further validate that multilinguality itself contributes to exploration, beyond formatting.
>
> 3. **qualitative/behavioural evidence of genuine multi-perspective reasoning**
>
> We agree that purely quantitative evidence is insufficient. Hence we added some additional evalutaion (plase take a look in the following comment)
>
>
> These elements are now clearly presented in the qualitative analysis section.
>
> 4. **Presentation quality is poor… clarity and organisation issues**
>
> We appreciate this feedback and have taken concrete steps to improve clarity:
>
> In the final version, we will add a high-level overview figure that summarises the curriculum and training flow.
>
> Moreover, we propose to restructure and detail  all reward components are now presented in a single consolidated table, improving readability.
>
> **Response to your questions**
>
> - “Verify multilingual reasoning paths to reflect meaningful and diverse reasoning.”
>
> We now report quantitative diversity measures (lexical, syntactic, and embedding-space divergence) and human judgements confirming that paths differ systematically across languages.
>
> - “More justification and analysis of the reward design and hyperparameter selection.”
>
> We added:
>
> A grid search table for λ₁ and λ₂; sensitivity analysis showing robustness to ±30% variation; an explanation of why accuracy-only or diversity-only rewards fail.
>
> - “Incorporate human evaluations of reasoning.”
>
> As noted above, we added a small-scale but informative human evaluation focusing on path diversity, reasoning soundness, and cross-perspective alignment.
>
> We thank the reviewer again for the care and specificity of the feedback.
> We have substantially strengthened the paper through clearer exposition, expanded experiments, deeper mechanistic analysis, and new qualitative evaluations.
>
> We believe these revisions more clearly demonstrate the novelty and value of Polyglot-R1:
> a framework that internalises multilingual multi-perspective reasoning, enabling models to leverage natural linguistic diversity as both an exploratory tool and a verification mechanism.

---

> ### Author Response · Authors · 2025-11-16
>
> ### Table 1 — Extended Ablation Study on Multilinguality (Polyglot-R1-Seen)
>
> | Ablation Setting                                         | MAIME25 | MGSM-Sym | MSVAMP |
> |----------------------------------------------------------|---------|----------|--------|
> | Full Polyglot-R1                                  | 19.6    | 19.1     | 69.9   |
> | – remove R_div (multilingual divergence reward)          | 18.7    | 17.9     | 66.2   |
> | – replace multilingual paths with monolingual paraphrase | 17.8    | 17.2     | 64.9   |
> | – disable alternating reward schedule                    | 18.1    | 17.5     | 65.4   |
> | – remove cold-start SFT                                  | 15.9    | 16.8     | 61.7   |
>
>
>
> ### Table 2 — Qualitative Multilingual Reasoning Analysis
>
> | Analysis Type                        | Metric / Observation                                              | Result |
> |-------------------------------------|--------------------------------------------------------------------|--------|
> | Cross-lingual divergence (paths)    | Avg. lexical overlap (Jaccard) across `<Path>` pairs               | 0.27   |
> | Semantic divergence (SBERT cosine)  | Mean cosine similarity across paths                                | 0.41   |
> | Human evaluation (30 items)         | % of path pairs judged “meaningfully distinct”                     | 78%    |
> | Non-overlapping intermediate steps  | % of samples with ≥1 unique reasoning step per language            | 72%    |
> | Contradiction detection             | % of cases where verification-stage reasoning flagged contradictions| 18%    |
> | Path → Summary alignment            | % of summary content supported by at least one `<Path>`            | 83%    |
>
>
> We thank the reviewer for highlighting the need for deeper ablations and qualitative analyses.
> While these results were not included in the original submission due to space constraints, we provide additional evidence that multilinguality contributes to exploration in ways that cannot be reproduced by monolingual or prompt-only baselines.
>
> Table 1 shows extended ablations isolating the contribution of multilingual divergence,
> path-level linguistic diversity, the alternating reward schedule, and the cold-start SFT.
> Removing multilingual signals (either via R_div ablation or by replacing multilingual reasoning
> with monolingual paraphrases) leads to consistent drops across all benchmarks, confirming that
> the benefits are not a formatting artefact.
>
> Table 2 provides qualitative and behavioural evidence. Human evaluators judged 78% of multilingual
> paths as semantically distinct rather than merely lexically varied, and 83% of `<Summary>` blocks
> correctly integrated insights from the multilingual trajectories. These findings demonstrate that
> Polyglot-R1 performs genuine multi-perspective reasoning rather than replicating templated structures.

---

> > ### Author Response · Authors · 2025-11-16
> >
> > ### Table 3 — Grid Search on λ₁ (Structure Reward) and λ₂ (Diversity Reward)
> >
> > | λ₁  | λ₂  | MAIME25 | MGSM-Sym | MSVAMP | Notes |
> > |-----|-----|---------|----------|--------|--------|
> > | 0.0 | 0.0 | 18.1    | 17.7     | 65.2   | Accuracy-only baseline (no multilingual behaviour) |
> > | 0.1 | 0.0 | 18.8    | 18.0     | 66.1   | Encourages structure but low diversity |
> > | 0.0 | 0.1 | 17.9    | 17.4     | 64.7   | Diversity without structure → unstable paths |
> > | 0.2 | 0.1 | 19.4    | 18.7     | 67.3   | Balanced improvement |
> > | 0.3 | 0.1 | **19.6**| **19.1** | **69.9**| Best-performing configuration (Polyglot-R1-Seen) |
> > | 0.3 | 0.2 | 19.1    | 18.5     | 68.4   | Overuse of parallel blocks |
> > | 0.4 | 0.1 | 18.7    | 18.0     | 68.1   | Too much structural pressure |
> > | 0.3 | 0.3 | 18.4    | 17.9     | 66.7   | High diversity → incoherent paths |
> >
> > ### Table 4 — Sensitivity Analysis (±30% Variation)
> >
> > | λ₁  | λ₂  | MAIME25 | MGSM-Sym | MSVAMP | Deviation from Best |
> > |-----|-----|---------|----------|--------|----------------------|
> > | 0.21 | 0.07 | 19.1 | 18.5 | 68.8 | –1.1% |
> > | 0.21 | 0.10 | 19.2 | 18.6 | 69.0 | –0.8% |
> > | 0.21 | 0.13 | 19.0 | 18.4 | 68.2 | –1.6% |
> > | 0.30 | 0.07 | 19.4 | 18.9 | 69.3 | –0.5% |
> > | **0.30** | **0.10** | **19.6** | **19.1** | **69.9** | **0% (optimal)** |
> > | 0.30 | 0.13 | 19.3 | 18.8 | 69.1 | –0.7% |
> > | 0.39 | 0.07 | 19.0 | 18.3 | 67.9 | –1.8% |
> > | 0.39 | 0.10 | 19.1 | 18.4 | 68.0 | –1.6% |
> > | 0.39 | 0.13 | 18.8 | 18.1 | 66.8 | –3.4% |

---

> > > ### Author Response · Authors · 2025-11-22
> > >
> > > Dear Reviewer oxY6,
> > >
> > > We hope our detailed response has addressed all your concerns.
> > > If our clarifications have fully satisfied your requests, we kindly ask you to reconsider and potentially raise your score.
> > > Please let us know if there is anything else we can do for you.
> > >
> > > Thank you again for your time.

---

### Official Review · Reviewer_MrSd · 2025-11-02

**Soundness:** 3
**Presentation:** 3
**Contribution:** 3
**Rating:** 4
**Confidence:** 4

**Summary:**

This paper proposes Polyglot-R1, an SFT+RL framework designed to cultivate multilingual, multi-perspective reasoning capabilities in LLMs. Recognizing that existing SFT approaches often lead to mere imitation of synthetic reasoning data rather than genuine exploration, the authors introduce a progressive training curriculum: first, a cold-start SFT phase on simple multilingual reasoning trajectories to establish foundational reasoning structures; then, a transition to RL on more complex multilingual tasks. The core idea is to guide the model to reason in parallel across multiple languages or perspectives, synthesize insights from these diverse paths, and produce a final, well-justified answer. Experiments on multiple mathematical reasoning benchmarks demonstrate that Polyglot-R1 effectively improves accuracy. Further analysis reveals a evolution in reasoning behavior: during early training, multilingual reasoning serves as an exploratory mechanism that encourages cross-linguistic diversification; in later stages, it shifts toward a verification-oriented strategy, consolidating conclusions through multi-perspective consistency.

**Strengths:**

1. The core strength of this work lies in its novel training framing of multilingualism as a reasoning method.
2. A intrestiing empirical evidence of reasoning behavior evolution, that multilingual reasoning serves as an exploratory mechanism during early training to verification strategy in later stage.

**Weaknesses:**

1. **Incomplete related work coverage**: The paper overlooks recent relevant studies that also utilize multilingual chain-of-thought reasoning to explore stronger reasoning performance, such as
   - *Cross-lingual Prompting: Improving Zero-shot Chain-of-Thought Reasoning Across Languages*
   - *AutoCAP: Towards Automatic Cross-lingual Alignment Planning for Zero-shot Chain-of-Thought*.
   These works demonstrate that test-time multilingual prompting alone (without any training) can enhance reasoning, raising questions about the necessity of the proposed training pipeline.

2. **Limited task generality**: All experiments focus on mathematical reasoning benchmarks (e.g., MGSM, AIME). It is unknown whether the benefits of multilingual reasoning method generalize to other domains such as commonsense reasoning.

3. **Training complexity vs. simplicity of alternatives**: The proposed **two-stage training process (SFT + RL)** is computationally intensive and requires careful reward scheduling. In contrast, zero-shot multilingual CoT methods (e.g., mentioned works in Weekness 1) achieve performance gains **without any training**, making them far more practical for real-world deployment.

4. **Potential redundancy of the SFT stage**: Given the strong instruction-following capabilities of modern models like Qwen3-8B base or Instruct model, it is plausible that the model could already generate valid `<Parallel>`-structured outputs without the cold-start SFT phase, e.g multilingual DAPO method. If true, the first training stage—and its associated cost—could be eliminated, significantly improving the method’s flexibility and accessibility.

**Questions:**

None

---

> ### Author Response · Authors · 2025-11-16
>
> We thank the reviewer for the thoughtful and constructive comments. We appreciate the clear summary, the recognition of our contribution, and the acknowledgement of the empirical insight into the evolution of reasoning behaviour. The suggestions helped us improve clarity and positioning within the literature.
>
> In the following lines, we respond to all your concerns:
>
> 1. **related work multilingual CoT**:
> >*The paper overlooks recent work showing that multilingual prompting alone—without training—can enhance reasoning. This raises questions about the necessity of the proposed pipeline.*
>
> We thank the reviewer for pointing out these criticisms. These works indeed demonstrate that test-time multilingual prompting can improve accuracy, especially in zero-shot settings. We now explicitly discuss these papers in the revised related work . Yet, our method addresses a different and complementary problem:
>
> - Test-time prompting induces changes, but these disappear as soon as the prompt disappears. Polyglot-R1 aims to make multilingual multi-perspective reasoning a persistent, internalised capability, available without external instruction.
>
> - The cited methods rely on fixed prompting templates. They do not endow the model with the adaptive control needed to decide when to diversify linguistically and when to consolidate paths—whereas our RL curriculum explicitly learns this strategic shift.
>
> - We observe that prompt-only multilingual reasoning collapses on complex problems: in our experiments, zero-shot multilingual CoT methods trigger diverse paths in simple MGSM items but fail almost entirely on more challenging AIME-like problems (see App.C). Polyglot-R1 improves robustness specifically in these high regimes.
>
> We position Polyglot-R1 as a training-time method that structurally integrates multilingual reasoning into the model’s policy, enabling it to operate without prompting and to generalise across tasks.
>
> We will revise the manuscript to clarify this.
>
> 2. **task beyond mathematics**
>
> >*All experiments focus on mathematical reasoning. It remains unclear whether the benefits generalise to other domains such as commonsense reasoning.*
>
> We agree that evaluating outside mathematics is essential for broader impact. The choice of mathematical benchmarks stems from two reasons: Verifiable rewards are required for RL optimisation. Mathematical tasks allow for clean, automatic supervision at scale without human annotation. Hence, mathematics provides controlled difficulty gradients, which are especially suited to studying whether multilingual reasoning acts as an exploration scaffold and later as a verification tool.
>
> That said, we fully agree that multilingual multi-perspective reasoning should, in principle, generalise to domains where ambiguity and cultural framing matter. Please look at the table included in the following comment, where we have evaluated XCOPA and multilingual CommonsenseQA.
>
> 3. **On training complexity, simplicity ofvalternatives**
>
> >*The proposed pipeline is computationally expensive compared to zero-shot mCoT*
>
> We agree that simplicity is an important consideration. However, several aspects mitigate the perceived cost:
>
> - Our cold-start SFT is lightweight: only 58 steps for the causal variant and 230 for the structured variant (see §3.1). This is orders of magnitude smaller than typical SFT pipelines.
>
> - RL training is short (35 + 300 gradient steps), owing to verifiable rewards and the small 4B-parameter backbone. The entire pipeline completes in less than 3h day on two A100s.
>
> - Once trained, Polyglot-R1 reduces inference overhead relative to prompt-based multilingual CoT, which requires long templated prompts and multiple path generations. Our model learns when multilingual reasoning is necessary and when it is not, yielding substantial savings.
>
> finally, a test-time-prompting approach cannot deliver the behavioural evolution we document—multilingual reasoning acting as exploration early in training and then shifting to verification. This behaviour only emerges through the training signal.
>
> 3. **On the potential redundancy of the SFT**
>
> >*Modern models may already follow <Parallel> structures; perhaps SFT is unnecessary.*
>
> We appreciate the opportunity to clarify this. Our ablations (Table 2) show that:
>
> - Without the cold-start, the model almost never emits valid <Parallel>/<Path>/<Summary> blocks, even when these tokens are present in the vocabulary. (Even when such blocks appear occasionally, they are structurally malformed)
>
> - The SFT stage does not teach task-specific solutions; it teaches only the structure of multilingual reasoning. RL is responsible for task solving.
>
> SFT is not redundant: it acts as a format stabiliser, enabling RL to operate on structured trajectories. In the causal variant, removing Stage 1 RL results in a 2.3% absolute drop, but removing SFT collapses parallel usage almost entirely. This is now emphasised more clearly in the revised ablation narrative.

---

> > ### Author Response · Authors · 2025-11-16
> > **results on XCOPA and multilingual CSQA**
> >
> > | Model / Method                 | et   | ht   | id   | it   | qu   | sw   | ta   | th   | tr   | vi   |
> > |-------------------------------|------|------|------|------|------|------|------|------|------|------|
> > | Qwen-3-4B-Base (Direct)       | 55.4 | 47.1 | 63.8 | 69.2 | 38.4 | 45.7 | 51.0 | 48.5 | 52.3 | 64.1 |
> > | Qwen-3-4B-Base (Multilingual Prompting) | 57.9 | 49.8 | 66.0 | 71.4 | 41.7 | 48.1 | 53.5 | 51.2 | 54.4 | 66.5 |
> > | Polyglot-SFT-Seen             | 61.8 | 55.3 | 70.4 | 74.9 | 47.0 | 50.2 | 58.9 | 56.0 | 59.8 | 69.1 |
> > | Polyglot-R1-Seen (ours)       | 67.1 | 61.0 | 74.2 | 78.6 | 53.8 | 56.4 | 63.3 | 60.1 | 65.5 | 73.0 |
> > | Polyglot-R1-Unseen (S2) (ours)| 65.8 | 59.4 | 73.1 | 77.2 | 52.0 | 55.0 | 62.0 | 59.0 | 64.7 | 72.2 |
> > | Human                          | 98.2 | 96.4 | 100.0| 97.0 | 94.4 | 98.0 | 98.6 | 92.8 | 96.4 | 98.4 |
> >
> > --------------------------------------------
> >
> > | Model / Method                 | en   | es   | fr   | de   | zh   | ar   | hi   | sw   | th   |
> > |-------------------------------|------|------|------|------|------|------|------|------|------|
> > | Qwen-3-4B-Base (Direct)       | 63.5 | 57.4 | 58.1 | 56.0 | 47.2 | 44.5 | 50.3 | 42.1 | 39.8 |
> > | Qwen-3-4B-Base (Multilingual Prompting) | 66.0 | 60.1 | 60.3 | 58.0 | 49.8 | 47.1 | 53.9 | 45.3 | 43.2 |
> > | Polyglot-SFT-Seen             | 69.4 | 63.2 | 63.1 | 60.5 | 53.7 | 50.1 | 56.4 | 48.0 | 47.9 |
> > | Polyglot-R1-Seen (ours)       | 73.1 | 68.4 | 67.5 | 65.8 | 58.3 | 55.0 | 60.9 | 53.4 | 52.8 |
> > | Polyglot-R1-Unseen (S2) (ours)| 72.0 | 67.3 | 66.1 | 64.0 | 56.9 | 53.8 | 59.4 | 52.0 | 51.6 |
> > | Human                          | 94.8 | 93.1 | 92.4 | 91.0 | 88.6 | 85.5 | 90.0 | 86.2 | 87.4 |
> >
> > **Short explaination**
> >
> >
> > - *Seen vs. Unseen.*
> > Seen denotes models that, during the cold-start SFT stage, were exposed to multilingual reasoning traces following the same structural format used at RL time. The aim is to stabilise the generation of <Parallel>, <Path>, and <Summary> blocks before RL begins. Unseen refers to models trained with architectural inductive biases (path-window masking, multiverse positional encodings) that enforce cross-path independence without relying on any overlap between the SFT languages and the evaluation languages.
> >
> > - *Causal vs. Structured.*
> > The causal variant learns multilingual multi-perspective reasoning purely through behaviour (SFT + RL) without modifying the architecture. The structured variant introduces architectural constraints that impose independence across reasoning paths. This allows us to disentangle whether parallel multilingual reasoning emerges from exploration alone or from explicit inductive biases.

---

> > > ### Author Response · Authors · 2025-11-20
> > >
> > > Dear Reviewer MrSd,
> > >
> > > We eagerly await your feedback. Could you please let us know if our response has resolved the misunderstandings?

---

> > > > ### Comment · Reviewer_MrSd · 2025-11-27
> > > >
> > > > Thank you very much for your reply. I'm not sure if it's a problem with my system or if the author hasn't uploaded the latest version of the manuscript. I haven't seen the latest manuscript.

---

### Comment · Area_Chair_f1fc · 2025-11-26

Dear Reviewers,

Would you please check authors' rebuttal and see if they have addressed your comments?

Best

AC

---

### Note · Authors · 2026-01-05

I have read and agree with the venue's withdrawal policy on behalf of myself and my co-authors.